# Resolution Limits in Photoacoustic Imaging Caused by Acoustic Attenuation

**DOI:** 10.3390/jimaging5010013

**Published:** 2019-01-10

**Authors:** Peter Burgholzer, Johannes Bauer-Marschallinger, Bernhard Reitinger, Thomas Berer

**Affiliations:** Research Center for Non Destructive Testing (RECENDT), Linz 4040, Austria

**Keywords:** photoacoustic imaging, ultrasound attenuation, inverse problem, image reconstruction

## Abstract

In conventional photoacoustic tomography, several effects contribute to the loss of resolution, such as the limited bandwidth and the finite size of the transducer, or the space-dependent speed of sound. They can all be compensated (in principle) technically or numerically. Frequency-dependent acoustic attenuation also limits spatial resolution by reducing the bandwidth of the photoacoustic signal, which can be numerically compensated only up to a theoretical limit given by thermodynamics. The entropy production, which is the dissipated energy of the acoustic wave divided by the temperature, turns out to be equal to the information loss, which cannot be compensated for by any reconstruction method. This is demonstrated for the propagation of planar acoustic waves in water, which are induced by short laser pulses and measured by piezoelectric acoustical transducers. It turns out that for water, where the acoustic attenuation is proportional to the squared frequency, the resolution limit is proportional to the square root of the distance and inversely proportional to the square root of the logarithm of the signal-to-noise ratio. The proposed method could be used in future work for media other than water, such as biological tissue, where acoustic attenuation has a different power-law frequency dependence.

## 1. Introduction

Photoacoustic tomography (PAT), which is also called optoacoustic or thermoacoustic tomography, is based on the generation of ultrasound following a temperature rise after the illumination of light-absorbing structures within a semitransparent and turbid material, such as a biological tissue. It provides optical images with specific absorption contrast [1,2,3]. Therefore, it offers greater specificity than conventional ultrasound imaging with the ability to detect hemoglobin, lipids, water, and other light-absorbing chromophores, but with greater penetration depth than purely optical imaging modalities that rely on ballistic photons. In PAT, the temporal evolution of the acoustic pressure field is sampled using an array of ultrasound detectors placed on the tissue surface or by moving a single detector across the detection surface. Images of the optical absorption within the tissue are then reconstructed by solving an inverse source problem [3,4,5].

In this work, the influence of acoustic attenuation on the achievable spatial resolution is investigated. At depths larger than the range of the ballistic photons, i.e., more than a few hundreds of microns in tissue, light is multiply scattered, and the spatial resolution is limited by acoustics. As higher acoustic frequencies, which have smaller wavelengths and allow a better resolution, are more strongly damped than lower frequencies, the spatial resolution decreases with depth. The spatial resolution is limited at such depths by the acoustic diffraction limit that corresponds to the highest detectable frequency. The ratio of the imaging depth to the best spatial resolution is roughly a constant of 200 [3]. Only recently published non-linear imaging methods, which use additional information such as the sparsity of the imaged structure, can overcome the acoustic diffraction limit, and are therefore called “super-resolution” [6,7]. Technical limitations, such as a bandwidth mismatch between the acoustic transducer and the acoustic signal on the sample surface or the noise of an amplifier, can reduce the resolution in addition, but can be avoided in principle.

There have been several attempts for mathematically compensating the acoustic attenuation to obtain images with a higher spatial resolution. Already in 2005, La Riviere et al. proposed an integral equation that related the measured acoustic signal at a given transducer location in the presence of attenuation to the ideal signal in the absence of attenuation [8,9]. Ammari et al. later gave a compact derivation of this integral equation directly through using the wave equations, which is valid for all of the dimensions [10]. Dean-Ben et al. described the effects of acoustic attenuation (amplitude reduction and signal broadening), compared these effects to the influence of the transducer bandwidth and space-dependent speed of sound, and established a correction term similar to La Riviere, but for space-dependent attenuation [11]. Kowar and Scherzer used a similar formulation for other lossy wave equations [12].

Burgholzer et al. have directly compensated for the attenuation in photoacoustic tomography by using a time reversal finite differences method with a lossy wave equation [4,13,14,15,16]. The time reversal of the attenuation term causes the acoustic waves in the finite differences model to grow, as they propagate back in time through the tissue. At each time step, the total acoustic energy is controlled by cutting high-frequency signals, which would otherwise grow too quickly. This approach was later extended by Treeby et al. to account for the general power law absorption behavior [17,18]. Inspired by attenuation compensation in seismology, Treeby proposed a new method for attenuation compensation in photoacoustic tomography using time-variant filtering [19].

All of these attempts have one thing in common: the compensation of the frequency-dependent attenuation is an ill-posed problem that needs regularization. The physical reason for this ill-posedness is thermodynamics: acoustic attenuation is an irreversible process and the entropy production, which is the dissipated energy of the attenuated acoustic wave divided by the temperature, is equal to the information loss for the reconstructed image [16]. This also limits the spatial resolution, which correlates with the information content of the reconstructed image. To reach this thermodynamic resolution limit for the compensation of acoustic attenuation, it is necessary to measure the broadband ultrasonic attenuation parameters of tissues or liquids very accurately [20], and evaluate the existing mathematical models in order to get an accurate description of the attenuation [21].

In Section 2, the experimental set-up to measure the laser-induced acoustic wave at a varying distance is described. Section 3 gives the theoretical resolution limit for the acoustic attenuated waves in water. In Section 4, the measured acoustic wave signals are presented, the acoustic attenuation is compensated, and the theoretical limits are verified. In Section 5, the results are summarized, and the extension to other materials, such as biological tissue, is discussed.

## 2. Experimental Set-Up

Figure 1 schematically presents the experimental set-up. An unfocused piezoelectric immersion transducer (V358-SU, Panametrics, Waltham, MA, USA) with a center frequency of 50.6 MHz and a −6 dB bandwidth of 81.2% is used to detect the ultrasonic waves. The active element of the transducer has a diameter of 6.35 mm. After being amplified by a pulser/receiver (5073PR-40-E, Olympus NDT Inc., Waltham, MA, USA; not shown in the figure), the received signals are sampled by a digital oscilloscope (DSO 5043, 300 MHz; Agilent Technologies Inc., Santa Clara, CA, USA; not shown in the figure). The plane waves are generated on basis of the laser ultrasound technique [22,23]. To this end, a short laser pulse is directed onto an opaque target. Thereby, a thin region under the illuminated spot is rapidly heated. The resulting thermoelastic expansion of the heated volume causes the emission of broadband ultrasonic waves. Thereby, a broader frequency spectrum can be achieved compared to ultrasonic generation with another piezoelectric transducer. The laser pulses are generated with a frequency-doubled Nd:YAG laser at wavelength of 532 nm, with a pulse duration of six ns, a repetition rate of 20 Hz, and a beam diameter of six mm. Different pulse energies, ranging from 12 mJ to 65 mJ, were used. The mechanical components are designed in order to minimize the axial tilting between the ultrasonic source and detector and errors regarding the length of the water path. More details on the set-up can be found in [20]. By using different distance bolts in the mechanical set-up, water paths of 7.65 mm, 10.65 mm, and 24.65 mm were realized. 

## 3. Compensation of Acoustic Attenuation and the Limits of Resolution

For a plane wave, a Dirac’s delta pulse δ after traveling a distance r in a liquid, such as water, the pressure p can be described as (e.g., [11]):(1)p(r,t)=12ππα0rexp(−(rc−t)24α0r),
in which $\alpha_{0}$ is the attenuation constant, t is the time, and c is the sound velocity. The amplitude of the pulse is decreased, and the width of the Gaussian pulse is increased by a factor of $\sqrt{\alpha_{0}r}$. Without attenuation, the ideal wave is:(2)$$p_{ideal}\left(r,t\right)=\delta \left(\frac{r}{c}-t\right),$$
where the shape of the pulse does not change. According to [11], for a one-dimensional plane wave, a relationship between the non-attenuated plane wave $p_{ideal}$ and the attenuated wave p can be established:(3)$$p\left(r,t\right)=p_{ideal}\left(r,t\right)⋆\frac{1}{2\pi }\sqrt{\frac{\pi }{\alpha_{0}r}}exp\left(-\frac{t^{2}}{4\alpha_{0}r}\right),$$
where ⋆ denotes the time convolution.

In the frequency domain, acoustic attenuation in water is described by a power-law dependence, as can be seen by the Fourier transformation of Equation (3):(4)$$\tilde{p}\left(r,\omega \right)=\int_{-\infty }^{\infty }p\left(r,t\right)exp\left(i\omega t\right)dt=exp\left(i\omega \frac{r}{c}\right)exp\left(-\alpha_{0}\omega^{2}r\right),$$
where the Fourier transformation of the time convolution is the product of the Fourier transformations. Acoustic attenuation in a liquid is proportional to the square of the frequency ω. The Kramers–Kronig relationship states that for a power-law with an exponent of two, no dispersion occurs, and therefore the sound velocity c does not depend of the frequency ω. This makes the equations simpler than those for other exponents, such as a linear frequency behavior of attenuation, which is often assumed for biological tissues [11]. However, despite these mathematical difficulties, the same arguments that are presented here can be used for any materials and mathematical models for acoustic attenuation.

The limit in spatial resolution caused by frequency-dependent attenuation can be compensated numerically only up to a certain limit given by thermodynamics. The entropy production, which is the dissipated energy of the acoustic wave divided by the temperature, is equal to the information loss, which cannot be compensated by any reconstruction method. The compensation of acoustic attenuation is according to Equation (3) a deconvolution to get $p_{ideal}$ from the measured pressure p, which is mathematically an ill-posed inverse problem. Regularization methods, such as the truncated singular value decomposition (T-SVD) method, can solve this inverse problem. Before actually inverting Equation (3), the physical argument of entropy production as the cause for the “ill-posed-ness” of this inverse problem is elaborated in a similar way as we could do already for heat diffusion [24,25].

Thermodynamic fluctuations are the reason for the entropy production, which reduces the available information about the subsurface structures in the measured surface data. They are extremely small for macroscopic samples, but are highly amplified due to the ill-posed problem of the inversion of Equation (4) in the frequency domain. The factor $exp\left(-\alpha_{0}\omega^{2}r\right)$ for higher frequencies ω gets extremely small; therefore, for the inversion, the factor $exp\left(+\alpha_{0}\omega^{2}r\right)$ gets very large, which is also the factor for the amplification of the fluctuations.

Non-equilibrium thermodynamics has made enormous progress over the last decade. Heating light absorbing structures with a laser pulse and observing the induced pressure wave is definitely a process in which the system state is far from equilibrium. One comprehensive letter about non-equilibrium thermodynamics, the second law, and the connection between entropy production and information loss was published in 2011 by M. Esposito and C. van den Broeck [26]. They gave a proof that for two different non-equilibrium states evolving to the same equilibrium state, the entropy production *Δ_i_S* during the evolution from one state to the other is equal to the information loss *ΔI* = *k_B_ΔD*, with the Boltzmann constant *k_B_* and *ΔD* being the difference of the Kullback–Leibler divergence *D*, which is also called the relative entropy of these states. *D* is a measure of how “far” a certain state is away from equilibrium, see e.g., [27]. The entropy production for macroscopic states with small fluctuations around equilibrium turns out to be a good approximation that is equal to the dissipated energy *ΔQ* divided by the mean temperature *T*, so *Δ_i_S = ΔQ/T = k_B_ ΔD* [24,25].

In the frequency domain, the information loss with increasing time can be described by a cut-off frequency $\omega_{cut}$, which is calculated similar to that which is done for heat diffusion in thermography [24,25]. Equation (4) shows that after some distance *r*, the acoustic attenuation reduces the amplitude of $\tilde{p}\left(r,\omega \right)$ by a factor of $exp\left(-\alpha_{0}\omega^{2}r\right)$. The energy of the acoustic wave with frequency ω is proportional to the square of the pressure amplitude: $\Delta Q=0.5\chi \Delta V\;{\left|\tilde{p}\left(r,\omega \right)\right|}^{2}=0.5\chi \Delta V\;exp\left(-2\alpha_{0}\omega^{2}r\right)$ can be found e.g., in Morse and Ingard [28], where $\chi =1/\left(c^{2}\rho \right)$ is the adiabatic compressibility with the density ρ, and ΔV is the measurement volume. Since $k_{B}T/\left(\chi \Delta V\right)$ corresponds to the variance of the pressure (e.g., from Landau and Lifshitz [29]), one gets for the information content, which is the negative entropy of each frequency component $\Delta S_{\omega }=\Delta Q/T=0.5\;k_{B}SNR^{2}exp\left(-2\alpha_{0}\omega^{2}r\right)$. Note, that the signal-to-noise ratio (*SNR*) at the distance r=0 is the reciprocal value of the square root of the variance of the pressure, as the signal amplitude in the frequency domain is normalized to one (Equation (4)). Now, the cut-off frequency $\omega_{cut}$ is defined so that the information content in this frequency component is so low that its distribution cannot be distinguished from the equilibrium distribution within a certain statistical error level (Chernoff–Stein Lemma, see e.g., [27]). This error level can be set such that the amplitude of $\tilde{p}\left(r,\omega \right)$ is damped below the noise level for frequencies higher than $\omega_{cut}$ [24]. Using this error level, the acoustic wave at a distance r cannot be distinguished from equilibrium if $\Delta S_{\omega_{cut}}$ gets less than $0.5\;k_{B}$, which gives:$$\mathrm{SNR}exp\left(-\alpha_{0}\omega_{cut}{}^{2}r\right)\;=\;1.$$
(5)$$\mathrm{or}\;\omega_{cut}=\sqrt{\frac{\ln\left(SNR\right)}{\alpha_{0}r}}.$$

For the spatial resolution in photoacoustic imaging, the width of the acoustic signal in the time domain is essential. A small width enables a high spatial resolution, which corresponds to a broad frequency bandwidth. If the frequency bandwidth is limited by thermodynamic fluctuations according to Equation (5), the spatial resolution limit according to Nyquist is half the wavelength at this frequency:(6)$$\delta_{resolution}=\frac{\pi }{\omega_{cut}}c=\pi c\sqrt{\frac{\alpha_{0}r}{\ln\left(SNR\right)}}.$$

For our experimental set-up described in Section 2, the acoustic pulse is measured by a piezoelectric transducer. Its impulse response T(t) is taken into account in the time domain by an additional convolution in Equation (3), or in Equation (4) by an additional factor $\tilde{T}\left(\omega \right)$ in the frequency domain. The transducer limits the frequency bandwidth. This technical limit can be overcome (in principle) by using better transducers with higher bandwidths. As shown in the next section, our used transducer can measure the frequency components only up to approximately 100 MHz, which is for smaller propagation distances r significantly lower than the cut-off frequency $\omega_{cut}$ from Equation (5). Therefore, it is expected that a degradation of the resolution $\delta_{resolution}$ with increasing distance r according to Equation (6) can only be found for higher distances, where $\omega_{cut}$ is in the order of—or lower—than 100 MHz.

## 4. Results

From the arrival time of the pulses at various distances, the sound velocity was determined by linear regression as c = 1498 m/s at a water temperature of 25.5 °C. Interpolation from the acoustic attenuation data of Litovitz and Davis for this temperature gives $\alpha_{0}=5.24\cdot 10^{-16}{\left(\frac{rad}{s}\right)}^{-2}m^{-1}$ [30].

### 4.1. Measured Signals and Frequency Spectra

Figure 2 shows the measured acoustic pressure as a function of time at a distance r of 7.65 mm, 10.65 mm, and 24.65 mm. A total of 32 measurements were averaged to get a signal-to-noise ratio (SNR) of approximately 570 instead of 100 for single measurements. The arrival time of the ideal acoustic wave was subtracted, which allows plotting all of the signals with the same time scale. The corresponding frequency spectra of the measured signals from Figure 2 are shown in Figure 3.

### 4.2. Compensation of Acoustic Attenuation Using the Truncated Singular Value Decomposition (T-SVD) Method

The pressure signals p(r,t) (Figure 2) are measured at a sampling rate of one GHz, which gives a time discretization of 1 ns. The same time discretization is used for $p_{ideal}\left(r,t\right)$, and the convolution with the Gaussian function in Equation (3) can be described as a matrix equation:(7)$$p_{r}=M_{r}p_{ideal},$$
where $p_{r}$ and $p_{ideal}$ are vectors and $M_{r}$ is the convolution matrix, which describes the influence of acoustic attenuation. $M_{r}$ cannot be inverted directly, but the pseudo-inverse matrix $invM_{r}$ can be calculated by the truncated singular value decomposition (T-SVD) method (see e.g., [31]). It turns out that the SVD of $M_{r}$ is its Fourier transformation (represented as multiplication with matrix F), which is shown in Equation (4):(8)$$M_{r}=F^{*}\;diag\left(exp\left(-\alpha_{0}\omega^{2}r\right)\right)F,$$
where $F^{*}$ is the complex conjugated of F, and $diag\left(exp\left(-\alpha_{0}\omega^{2}r\right)\right)$ represents the diagonal matrix of the singular values. The truncation criterion for the T-SVD method is that the singular value gets smaller than *1/SNR* [31]. This gives exactly the same truncation value as the cut-off value in Equation (5), which was derived by setting the entropy production equal to the information loss. The pseudo-inverse matrix is:(9)$$invM_{r}=\;F^{*}diag\left(1,\ldots ,exp\left(+\alpha_{0}\omega_{cut}{}^{2}r\right),0,\ldots ,0\right)F,$$
where in the diagonal matrix, the reciprocal values of singular values are taken, and for singular values smaller than *1/SNR*, the value in the diagonal is set to zero. The ideal wave can be reconstructed by:(10)$$p_{ideal\_rec}=invM_{r}p_{r}.$$

According to Equation (4), the reconstruction of the ideal wave is the inverse Fourier transformation, where the frequency integral is taken from $-\omega_{cut}$ to $+\omega_{cut}$:(11)$$p_{ideal\_rec}\left(r,t\right)=\frac{1}{2\pi }\int_{-\omega_{cut}}^{+\omega_{cut}}exp\left(i\omega \frac{r}{c}\right)exp\left(-i\omega t\right)d\omega =\frac{1}{\pi }\frac{1}{t-\frac{r}{c}}sin\left(\omega_{cut}\left(\frac{r}{c}-t\right)\right).$$

This is a sinc function, where the maximum at a distance r is at $t=\frac{r}{c}$, which is the arrival time of the ideal wave (Equation (2)). The zero points are at:(12)$$t=\frac{r}{c}\pm \frac{\pi }{\omega_{cut}}=\frac{r\pm \delta_{resolution}}{c},$$
with $\delta_{resolution}$ from Equation (6).

For the measured signals (SNR = 570), the truncation frequencies from Equation (5) are 200 MHz, 170 MHz, and 112 MHz for a distance r of 7.65 mm, 10.65 mm, and 24.65 mm, respectively (Table 1). This would give a spatial resolution of 3.7 μm, 4.4 μm, and 6.7 μm for the best reconstructions according to Equation (6), as shown in Table 2.

In the used experimental set-up, the bandwidth of the transducer significantly influences that resolution. The truncation frequency of the used transducer is 125 MHz (Table 1). This is the frequency at which the signal gets lower than the noise level because of the band limitation of the transducer and the amplifier. At a distance of 7.65 mm and 10.65 mm, the truncation frequency from wave propagation is significantly higher (Table 1), and therefore, the truncation frequency of 125 MHz and the spatial resolution of six μm does not change (Table 2). After a traveling distance of 24.65 mm, the truncation frequency is reduced to 93 MHz, and the resolution degrades to approximately eight μm.

Figure 4 shows the reconstruction of the ideal signal according to Equation (10) from the measured pressure signals that are shown in Figure 2. As shown in Equation (11), these sinc functions have their maximum at time zero, as the time scale was shifted by $-\frac{r}{c}$ for each signal, which allows plotting all of the signals on the same time scale. The zero points are at $\frac{\pm \delta_{resolution}}{c}$ (see Equation (12)).

## 5. Discussion, Conclusions, and Outlook

The numerical compensation of acoustic attenuation is an ill-posed inverse problem that needs regularization. Here, for the reconstruction of the ideal waves without acoustic attenuation from the measured pressure data of plane waves at a varying distance, the truncated singular value decomposition (T-SVD) method in the frequency domain was applied for regularization.

Choosing an adequate regularization parameter, which is for the T-SVD method the truncation frequency $\omega_{cut}$, is essential. If this frequency is too high, reconstruction artifacts come up due to the amplification of fluctuations (noise), as shown in Figure 5 (dashed line). If it is taken too low, resolution is lost (Figure 5, dotted line). A physical argument from thermodynamics for choosing the regularization parameter could be given. For a certain propagation distance r of the acoustic wave in water, the truncation frequency $\omega_{cut}$ is the frequency where the acoustic wave is just damped to the noise level. Lower frequencies are damped less, and therefore, these frequency components can contribute to the reconstruction of the ideal wave. Higher frequencies are damped below the noise level, and therefore, they are truncated and cannot contribute to the reconstruction of the ideal wave.

It was shown that for this truncation frequency $\omega_{cut}$, the absorbed acoustic energy divided by the temperature, which is the entropy production for this frequency component, is equal to the information loss described by the relative entropy (Kullback–Leibler divergence *D*). The close connection of entropy production (dissipation) to the signal noise (fluctuation) is an example of the fluctuation–dissipation theorem in non-equilibrium thermodynamics [24,25]. This enables us to give a physical argument for the choice of the regularization parameter $\omega_{cut}$: either when the frequency components of the acoustic signal are just damped to the noise level, or, which is equivalent according to the fluctuation–dissipation theorem, where the information content gets smaller than a certain level, and then this component cannot be distinguished from thermodynamic equilibrium (Chernoff–Stein–Lemma).

For the chosen transducer with a truncation frequency of 125 MHz, it turns out that the acoustic attenuation in water is too low to detect a significant degradation in resolution for a propagation distance up to 20 mm, where according to Equation (5), the truncation frequency only for acoustic attenuation is equal to the truncation frequency of the transducer. This is different for a propagation distance of 24.65 mm. Then, the truncation frequency for acoustic attenuation is 112 MHz, and in combination with the transducer, it is 93 MHz (Table 1). For future work, either a transducer with a higher bandwidth or a liquid with higher attenuation, such as glycerin, may be used. The proposed method could be used for media other than water, such as biological tissue where acoustic attenuation is higher and has a different power-law frequency dependence. In a first approximation, the change of the sound velocity with frequency, which is called dispersion, could be neglected compared to attenuation [11]. Acoustic attenuation in biological tissue is caused by a combination of dissipation and scattering, and the power law describing the frequency-dependent attenuation should describe both dissipation and scattering. If the frequency dependence of scattering and dissipation is significantly different than that in the observed frequency range, the power law needs to be modified.

In this manuscript, only one-dimensional wave propagation in a plane wave model is presented. The results only refer to the axial resolution in the direction of propagation. The relationship between the non-attenuated plane wave $p_{ideal}$ and the attenuated wave p as shown in Equation (3) is local, which means that for $p_{ideal}$ and for p, the same location in the sample is used, and the relation is the same for one, two, or three-dimensional wave propagation [10]. We have shown this explicitly by simulating the attenuated signal at a distance of 10 mm from a 200-micron thick layer (1D), a cylinder (2D), and a sphere (3D) as an acoustic source. Reconstruction in the axial direction was the same [14,15]. Therefore, it is sufficient to describe plane wave propagation. Of course, the resolution in higher dimensions varies for different directions. For one-dimensional propagation, no reconstruction is necessary, because the resolution is directly given by $p_{ideal\_rec}$ in Equation (11). For two or three-dimensional wave propagation, reconstruction methods such as time reversal reconstruction [4] can be used for image reconstruction from $p_{ideal}$. For lateral reconstruction perpendicular to the axial direction, propagation directions that have an angle θ to the axial direction are necessary. For such propagation directions, the distance to the sample surface increases with r/cos(θ), which especially broadens the resolution in the lateral direction.

## Figures and Tables

**Figure 1 jimaging-05-00013-f001:**
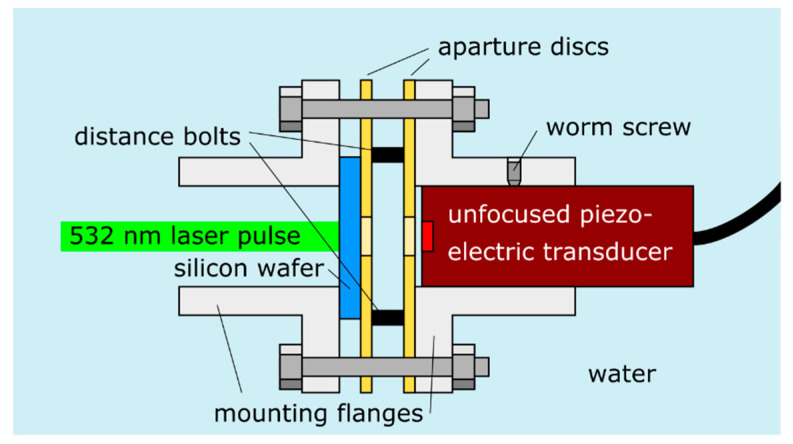
Experimental set-up for the generation and measuring of acoustic plane waves in water. Short laser pulses are directed on a silicon wafer. Fast local heating due to optical absorption leads to the emission of ultrasonic waves. The ultrasound detector is held by a mounting flange, and is axially fixated with a worm screw. The length of the water path between the silicon wafer (i.e., the ultrasound emitter) and the detector can be adjusted with distance bolts between two aperture discs. The silicon wafer is pressed onto one aperture disc by another mounting flange, which is bolted to the opposite flange.

**Figure 2 jimaging-05-00013-f002:**
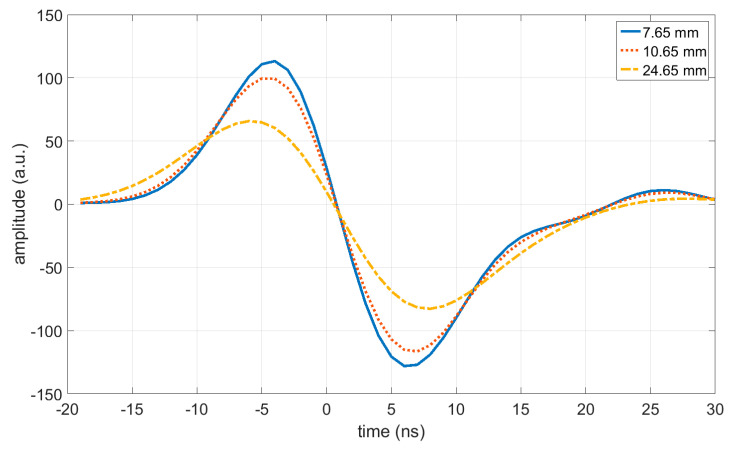
Measured acoustic pressure as a function of time at a distance *r* of 7.65 mm, 10.65 mm, and 24.65 mm. A total of 32 measurements were averaged to get a signal-to-noise ratio (SNR) of approximately 570 instead of 100 for single measurements. The time scale was shifted for each signal, which allows plotting all of the signals with the same time scale.

**Figure 3 jimaging-05-00013-f003:**
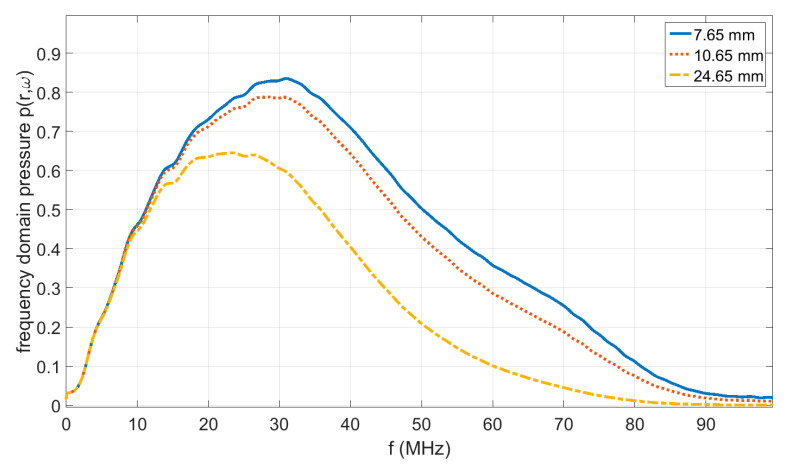
Amplitude of the Fourier transformation in time to get the frequency domain pressure amplitude from the measured signals shown in Figure 2 at a distance *r* of 7.65 mm, 10.65 mm, and 24.65 mm. The frequency response of the transducer is taken into account in the frequency domain by an additional factor $\tilde{T}\left(\omega \right)$ in Equation (4). The transducer limits the frequency bandwidth, and therefore, a significant influence according to Equation (4) on the bandwidth of the signals can be seen only for larger distances, such as that at 24.65 mm.

**Figure 4 jimaging-05-00013-f004:**
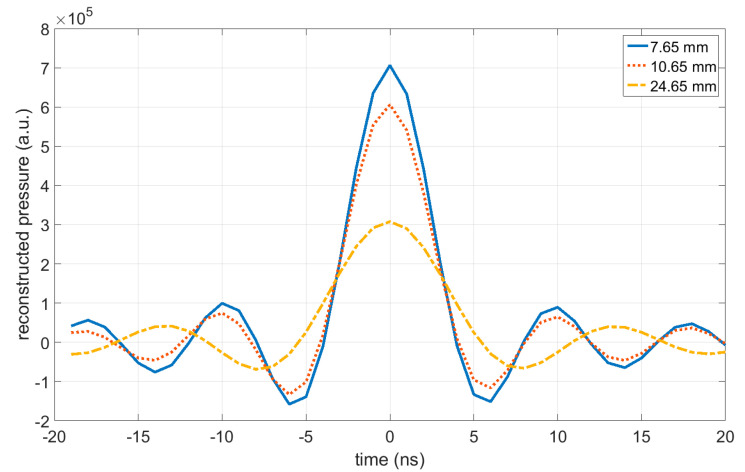
Reconstructed ideal waves according to Equation (10) from the measured acoustic pressure signals shown in Figure 2. According to Equation (11), these sinc functions have their maximum at time zero (the time scale was shifted for each signal, which allows plotting all of the signals on the same time scale) and the zero points are at $\frac{\pm \delta_{resolution}}{c}$.

**Figure 5 jimaging-05-00013-f005:**
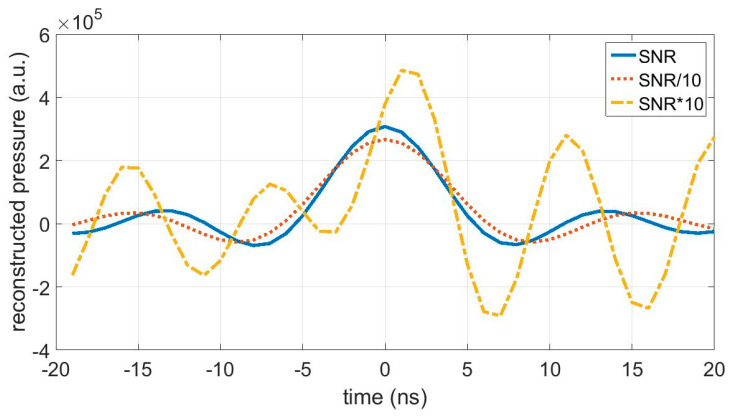
Reconstructed ideal waves for the signal at a distance 24.65 mm where the signal-to-noise ratio (SNR) of 570 for the reconstruction is divided by 10 (dotted line) or multiplied by 10 (dashed line). The width of the reconstruction gets higher for a lower SNR, because the truncation frequency is reduced. The truncation frequency is shifted from 93 MHz (Table 1) to 80 MHz, and the spatial resolution degrades from 8.1 μm (Table 2) to 9.3 μm. If a higher SNR is assumed than that provided by the experimental data, large artifacts caused by the high amplification of the fluctuations appear. In that case, proper reconstruction is not possible (dashed line). By such a SNR, the truncation frequency would be shifted from 93 MHz to 114 MHz, and the spatial resolution would be enhanced from 8.1 μm to 6.6 μm.

**Table 1 jimaging-05-00013-t001:** Truncation frequency $\omega_{cut}/\left(2\pi \right)$ in MHz.

Propagation Distance	Only for Acoustic Attenuation	Taking Transducer Bandwidth into Account
7.65 mm	200	125
10.65 mm	170	125
24.65 mm	112	93

**Table 2 jimaging-05-00013-t002:** Spatial resolution $\delta_{resolution}$ in μm.

Propagation Distance	Only for Acoustic Attenuation	Taking Transducer Bandwidth into Account
7.65 mm	3.7	6
10.65 mm	4.4	6
24.65 mm	6.7	8.1

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
