# Peer review of "Resolution Limits in Photoacoustic Imaging Caused by Acoustic Attenuation"

_2313-433X, 2019, doi:10.3390/jimaging5010013_

Round 1

Reviewer 1 Report

                In the manuscript titled “ Resolution limit in photoacoustic imaging caused by acoustic attenuation” the authors attempt to investigate the influence of acoustic attenuation on achievable spatial resolution. A simple experimental setup was constructed similar to authors’ previous work in reference 20 in which acoustic waves generated photo acoustically were measured using a transducer at varying path lengths. In section 3 of the manuscript a theoretical calculation of the entropy driven resolution limit. The results show how the measured attenuation of water and the resulting limit of resolution agrees with the theoretical limit when transducer response is taken into account.

 Comments:  I commend the authors on a well written article. As the authors point out, precise determination of acoustic attenuation in tissues and other media under imaging is critical for the reconstruction algorithms. The experimental verification of the thermodynamically limited resolution is also of scholarly interest.  

On the other hand, as the authors themselves seem to admit, the experiment carried out in this manuscript was severely limited in scope. Only one of the path lengths (24.65 mm) does even come close to something being limited by attenuation. Even in that case bandwidth of the transducer has a significant effect. The experiment could have easily done by either substantially increasing path lengths under study or using liquids with higher attenuation.  

In the hope that this article will inspire the authors or others to attempt further studies, I recommend publication as such.

Reviewer 2 Report

In this work, the authors used water as the simple acoustic attenuation model to study resolution limit in photoacoustic imaging caused by acoustic attenuation and its directly related parameters (e.g. frequency, spatial distance, and SNR) and further quantitatively compensated the attenuated signal using a truncated singular value decomposition method. Their results were derived from theoretical wave equations and then validated in experiments. This work is interesting and could provide new insight about photoacoustic image resolution limit. It can be considered for publication after addressing minor concerns listed as below:

1.      In their designed experiment, they used water as the acoustic attenuation medium, which is the simplest case in photoacoustic imaging and the acoustic attenuation is only caused by thermal assumption. But in real biological tissue imaging, due to tissue heterogeneity, acoustic scattering becomes another major source that causes the acoustic attenuation. Which factor dominates in real tissue imaging and how they play a role in resolution? The author should discuss this factor and the potential challenges in applying their method for real tissue imaging.

2.      The authors used a plane wave model and presented the time-trace signal measured in their experiment. So I assume they specifically refers axial resolution. For lateral resolution, are these conclusions still hold? There should be signal broadening in both axial and lateral directions. Please clarify/discuss? 

3.      There are a few steps missing or unclear in deriving equation for resolution limit, so it makes readers hard to follow, e.g. how did you get the equation in Line 159? How did you get the equation in Line 165?

4.      Fig. 5a and Fig. 5b can be combined into one figure. And The figure label is overlaid with the figure content in Fig. 5a. Please correct.

Round 2

Reviewer 2 Report

In the revised manuscript, the authors have addressed all my concerns. This manuscript now can be accepted for publication. A few minor edits are further suggested:

The authors should be clear in the manuscript that acoustic attention is caused by combination of dissipation and scattering instead of pure thermal assumption.

Typos still found in their revision, e.g “power low” and “pow-law”.

How you get SNR involved in equation in line 164 is still not clear, at least to me. SNR is supposed to be a measured value. Why you can directly put a measurement data into a theoretical derivation. Both SNR and resolution is highly dependent on the ultrasound detector you use. Please clarify.
